# Cost-Effective Photoacoustic Imaging Using High-Power Light-Emitting Diodes Driven by an Avalanche Oscillator

**DOI:** 10.3390/s25061643

**Published:** 2025-03-07

**Authors:** Alberto Prud’homme, Frederic Nabki

**Affiliations:** Department of Electrical Engineering, École de Technologie Supérieure, Montreal, QC H3C 1K3, Canada

**Keywords:** photoacoustic imaging (PAI), avalanche oscillator, photoacoustic effect, ultrasonic sensors, portable imaging systems, acoustic wave propagation

## Abstract

Photoacoustic imaging (PAI) is an emerging modality that merges optical and ultrasound imaging to provide high-resolution and functional insights into biological tissues. This technique leverages the photoacoustic effect, where tissue absorbs pulsed laser light, generating acoustic waves that are captured to reconstruct images. While lasers have traditionally been the light source for PAI, their high cost and complexity drive interest towards alternative sources like light-emitting diodes (LEDs). This study evaluates the feasibility of using an avalanche oscillator to drive high-power LEDs in a basic photoacoustic imaging system. An avalanche oscillator, utilizing semiconductor avalanche breakdown to produce high-voltage pulses, powers LEDs to generate short, high-intensity light pulses. The system incorporates an LED array, an ultrasonic transducer, and an amplifier for signal detection. Key findings include the successful generation of short light pulses with sufficient intensity to excite materials and the system’s capability to produce detectable photoacoustic signals in both air and water environments. While LEDs demonstrate cost-effectiveness and portability advantages, challenges such as lower power and broader spectral bandwidth compared to lasers are noted. The results affirm that LED-based photoacoustic systems, though currently less advanced than laser-based systems, present a promising direction for affordable and portable imaging technologies.

## 1. Introduction

Medical imaging plays a crucial role in diagnosing and monitoring various diseases and conditions. One promising modality that has gained significant attention in recent years is photoacoustic imaging, which combines the advantages of both optical imaging and ultrasound to provide high-resolution and functional information about biological tissues.

The photoacoustic effect, also known as the optoacoustic or thermoacoustic effect, has gained attention in medical imaging due to its ability to provide insights into tissue morphology, function, and composition. This effect involves generating acoustic waves through laser light absorption in tissues [1].

Photoacoustic imaging (PAI) combines the strengths of optical and ultrasound imaging. It offers high optical contrast by utilizing selective laser light absorption [2]. This feature enables the visualization of tissue structures with excellent resolution, surpassing the limitations of pure optical imaging. Furthermore, photoacoustic imaging allows deep tissue penetration, making it suitable for imaging organs like the breast, skin, brain, and prostate [1].

The PAI mechanism initiates with the delivery of pulsed laser light into the tissue of interest, resulting in selective optical absorption. This absorption induces rapid thermoelastic expansion and the subsequent generation of broadband ultrasonic waves. These photoacoustic resultant signals are subsequently captured by ultrasound transducers and converted into electrical signals for the computational reconstruction of detailed images [3].

In PAI, lasers serve as the primary light source for tissue excitation. The choice of laser parameters, including wavelength, pulse duration, and energy, significantly impacts the imaging outcomes. The selection of laser wavelength depends on the desired imaging depth and the absorption characteristics of the target tissue. For instance, near-infrared wavelengths between 700 and 1300 nm are commonly employed due to their deeper tissue penetration and absorption by endogenous chromophores like hemoglobin [4]. The pulse duration and energy levels of the laser pulse affect the photoacoustic signal strength, resolution, and potential tissue damage. Shorter pulse durations, ranging from 10 ns to a few hundred nanoseconds, are desirable for higher-resolution imaging [5].

Advancements in laser technology have led to the development of novel laser sources specifically designed for PAI. For instance, optical parametric oscillators (OPOs) offer tunable laser outputs, enabling multispectral imaging for the characterization of different tissue components based on their unique absorption spectra [6]. Fiber-based laser systems have also emerged, offering compact and flexible designs that facilitate the integration of photoacoustic imaging into various clinical settings [7].

Despite the benefits of using laser systems as excitation sources, their acquisition, maintenance, and operational costs are extremely high. Additionally, power consumption and equipment dimensions complicate the creation of portable systems, which has created a new tendency towards using alternative sources of excitation.

The utilization of light-emitting diodes (LEDs) as a light source in photoacoustic imaging systems offers several notable benefits and considerations. These characteristics contribute to the growing interest in and exploration of various imaging applications.

One significant advantage of LEDs is their cost-effectiveness compared to traditional laser sources. LEDs are relatively inexpensive, making them more accessible for research laboratories and clinical settings [8]. Furthermore, LEDs have a compact and lightweight design, allowing for the development of portable and handheld photoacoustic imaging systems. Their portability and ease of use make LEDs suitable for point-of-care applications and imaging in resource-limited settings.

LEDs also offer flexibility in wavelength selection. By using different LED configurations or employing LEDs with varying peak wavelengths, a wide range of wavelengths can be achieved for multispectral imaging. This enables researchers to explore different optical absorption properties and enhance tissue characterization [4].

Another advantage of LEDs is their long operational lifetime. LEDs have a significantly longer lifespan compared to lasers, reducing the need for frequent replacements and maintenance. This characteristic contributes to the cost-effectiveness of LED-based photoacoustic imaging systems.

However, there are limitations associated with using LEDs in photoacoustic imaging. LEDs typically have lower power radiation compared to lasers, which can result in reduced signal-to-noise ratios and imaging depths. The limited power output restricts the imaging of deep-seated tissues or structures [8]. Moreover, the spectral bandwidth of LEDs is broader compared to lasers, potentially affecting imaging specificity and the ability to differentiate different tissue components [9].

Light source control typically involves two key aspects: light source control and synchronization with the acoustic detection system. Light source control includes features such as precise current control, modulation capability, and synchronization inputs for external triggering. In the case of lasers, various types can be utilized, such as Q-switched Nd:YAG lasers, optical parametric oscillators, or tunable dye lasers, depending on the specific imaging requirements [10,11]. These laser sources offer the desired characteristics in terms of wavelength, pulse duration, and energy output.

Synchronization between the light and the acoustic detection system is crucial for accurate image acquisition. Timing and triggering circuits ensure precise synchronization between the light pulses and the detection system, enabling efficient capture of the photoacoustic signals [12]. This synchronization allows for the proper temporal alignment of the light excitation and the subsequent acoustic wave detection.

Additionally, modulation techniques can be employed to enhance imaging capabilities. Frequency-modulated continuous wave (FMCW) laser systems, for instance, enable depth-resolved imaging by sweeping the laser frequency and detecting the corresponding photoacoustic responses at different depths [13].

The feasibility of low-power portable photoacoustic systems requires the replacement of complex light source control systems, which must be capable of generating electrical pulses with durations in the nanosecond range and capable of obtaining the maximum amount of light from the LEDs, for which high voltages and currents are used. An affordable, low-cost option capable of delivering the necessary performance is an avalanche oscillator.

The avalanche oscillator (AO), also known as the avalanche transit-time oscillator (ATT), has potential applications in photoacoustic imaging when used with LEDs. By leveraging the principles of avalanche breakdown in semiconductor devices, the AO can generate high-voltage pulses suitable for driving LEDs in photoacoustic imaging systems.

In these applications, a reverse-biased p-n junction diode or a bipolar transistor serves as the active device in the AO circuit. When a reverse voltage is applied, the device operates in the reverse-biased breakdown region, leading to avalanche breakdown and the rapid generation of high-voltage pulses. By appropriately designing the device geometry, selecting the bias voltage, and optimizing other parameters, the AO can produce pulses at frequencies suitable for driving LEDs used in photoacoustic imaging.

The purpose of this work is to evaluate the feasibility and characterization of an avalanche oscillator exciting a high-power LED to be implemented as a basic photoacoustic imaging system, establishing the basis for future optimization work for a cost-effective portable photoacoustic imaging system.

## 2. Materials and Methods

This work comprises three primary components, which must function synergistically to establish the photoacoustic imaging system. The first component involves an avalanche oscillator responsible for generating pulses to drive the LED array. The second component includes an amplifier for the ultrasonic transducer, which facilitates the detection of the ultrasonic echo produced by the sample. Finally, the third component encompasses the characterization setup, designed to conduct comprehensive testing under specific conditions, thereby validating the anticipated outcomes.

An avalanche oscillator is an electronic circuit that exploits the avalanche breakdown phenomenon in semiconductor devices to generate high-frequency oscillations. This phenomenon occurs when a high voltage is applied to a semiconductor junction, causing a rapid and uncontrolled increase in current as electrons gain sufficient energy to ionize atoms in the lattice, leading to a chain reaction of generated electron–hole pairs. This highly non-linear process can produce oscillations in the microwave frequency range, often reaching several gigahertz.

The circuit typically includes a transistor or diode biased beyond its breakdown voltage, along with resonant circuits to stabilize the frequency of oscillation. The abrupt nature of the avalanche breakdown results in sharp current pulses, which, when coupled with a resonant LC circuit or a strip line resonator, can sustain oscillations at very high frequencies [14].

Avalanche oscillators are critical in applications requiring stable and precise high-frequency signals. They are commonly used in microwave communication systems, radar equipment, and signal generation for test and measurement instruments. One of the primary advantages of avalanche oscillators is their ability to achieve higher frequencies than many conventional oscillators, making them valuable in advanced electronic and communication technologies. Moreover, their simplicity and ability to generate large power outputs with minimal external components make them an attractive option for high-frequency circuit design [15].

The avalanche oscillator in Figure 1 was designed utilizing the Diodes Incorporated FMMT413TD avalanche transistor (Diodes Incorporated, Plano, TX, USA), which has a peak current of 50A and a breakdown voltage exceeding 150 V. The oscillator is powered by a 150 V DC source GW INSTEK GPE-2323 (GW INSTEK, Taipei, Taiwan), which is just below the breakdown level. The choice of passive components was made in accordance with the manufacturer’s guidelines to achieve a pulse of up to 50 A and a duration in the range of a few nanoseconds.

In the avalanche circuit used for this work, the resistor R_2_ plays a crucial role in regulating the current that charges capacitors C_1_ and C_2_, so the voltage across the capacitors gradually increases. This process continues until the voltage reaches the breakdown voltage of the transistor. At this point, the transistor enters the avalanche breakdown region, resulting in a rapid discharge of the capacitors. This discharge generates a pulse with a duration on the order of a few nanoseconds, characterized by an extremely high current in this case of 50 A.

In this work, the actual transistor breakdown voltage, approximately 160 V, was intentionally not used. Instead, an alternative method was employed to induce discharge by applying an external pulse to the base of the transistor. This pulse initiates the quick discharge in the transistor for the voltage across the capacitors to reach the LED.

In order to control the oscillation frequency, a function generator (RIGOL DG1022, Rigol, Beijing, China) producing a 1 V square wave with a 5% duty cycle is employed. This setup effectively triggers the discharge of the capacitors. The signal from the function generator was filtered and biased using components C_3_ and R_1_, resulting in a stable avalanche oscillator with uniform pulse amplitude and period. By adjusting the frequency of the base pulses, the postprocessing of the signal was significantly facilitated, reducing the noise factor during the self-oscillation of the closed-loop avalanche transistor.

The characterization of the oscillator was conducted using a RIGOL MS05104 oscilloscope (Rigol, Beijing, China). The tests presented in this work were conducted using a configuration with a 100 MHz bandwidth and an 8 GSa/s sampling rate, providing nanosecond-range resolution.

The array of LEDs can include multiple devices to increase the body excitation area during exploration. This approach has been demonstrated in other studies to enhance performance [16]. However, for the characterization phase of this study, only one LED was utilized, enabling the full luminous capacity to be focused on small areas and reducing the impact of disparities among the physical properties of each LED.

The LED used was the OSRAM SBRM4.24-V6V9-1-1-700-R33 (OSRAM, Kyoto, Japan), with a central wavelength of 660 nm. Each LED has a Luminous Flux of 1040 mW, with a forward voltage potential of 2 V and a current of 1.4 A. The wavelength employed in the LED can span a wide range; nevertheless, 660 nm was selected owing to its capability to excite both organic and synthetic materials, thereby affording greater versatility during the preliminary stages of this study. Subsequently, LEDs having different wavelengths can be substituted without compromising the integrity of the system [17,18].

In order to effectively capture the ultrasonic signal emitted after the light pulse excitation, the essential components encompass an ultrasonic transducer, in conjunction with a high-gain, low-noise amplifier that is required. Figure 2 depicts the setup used in this work, which includes a transimpedance amplifier (TIA) OPA4684 from Texas Instruments (Dallas, TX, USA) as the first stage, followed by a series-connected inverting amplifier powered by a symmetric power supply of +/−2.5 V.

A critical component in the reception stage is the ultrasonic transducer, where the central frequency, directionality type, and bandwidth impact its sensitivity, affecting resolution and maximum detection depth. The most commonly used transducers range in frequency from 1 MHz to 20 MHz, with higher frequencies providing greater resolution but reduced depth capability [19,20]. The ultrasonic transducer used in this work was the H2KLPY1100 from Unictron (Taipei, Taiwan), with a central frequency of 2.4 MHz and a linear pattern perpendicular to the front face of the device.

One of the significant limiting factors in the application of devices employing LEDs as light sources lies in the inherent rising and dropping times characteristic of the LED.

While it is feasible to measure the rise time and the duration of the electrical pulse with a high degree of accuracy using conventional electronic testing methodologies and an oscilloscope, determining the exact duration of the resultant optical pulse presents a more complex challenge.

Figure 3a shows the diagram of the setup used for measuring the duration of the light pulse generated by the LED. This was carried out using another LED of the same model placed inside a black nylon tube to prevent external light contamination. Using the same LED model ensures that the sensitivity of the receiving device is within the same range as the emitting source. In Figure 3b, the same setup is shown but with a black nylon plastic body between the two LEDs, preventing light from passing through the tube. By comparing measurements taken with and without the optical blockage, it becomes possible to directly evaluate and isolate the system’s response, ensuring the signal originates solely from the optical excitation of the LED. For the measurement of the light pulse, an oscilloscope was used with the LED directly connected with a 5-ohm resistor in parallel, ensuring rapid discharge after LED excitation. The resistor value was carefully optimized through extensive testing to prevent any influence on the measured pulse duration. The selected low resistance minimized the RC time constant of the circuit, thereby preserving the rise and fall times of the pulse. Although the resistor affected the pulse amplitude, its contribution to the RC time constant was negligible. It is important to note that the LED’s inherent capacitance can extend pulse duration when combined with higher resistance values due to an increased RC time constant. By choosing a 5-ohm resistor, the impact of the LED’s capacitance was effectively minimized, maintaining the temporal characteristics of the light pulse.

Once the pulse duration was confirmed, it was essential to evaluate whether the light pulse’s intensity and duration were sufficient to excite the material and induce the generation of an ultrasonic signal. This assessment involved verifying that the material’s response to the light pulse was adequate and that the generated ultrasonic signal was of sufficient amplitude. Furthermore, it was necessary to determine whether the amplifier could adequately amplify the ultrasonic signal for effective detection and analysis by the oscilloscope. This evaluation ensured the proper operation and reliability of the entire system, from light pulse generation to ultrasonic signal detection.

For the experiment, a black polyethylene membrane was selected for evaluation due to its high absorption of the light spectrum emitted by the LED, making it an appropriate material for this test [21]. Figure 4 illustrates the test setup, which consisted of a white PETG plastic casing fabricated using 3D printing. The LED was positioned on one side, oriented toward the centroid of the casing, while an Electret microphone (model CMEJ-0627-42-SP from CUI DEVICES, Lake Oswego, OR, USA) was placed on the opposite side. The microphone was directly connected to the TIA and subsequently to the oscilloscope.

The membrane was positioned equidistantly between the LED and the microphone, with a distance of 4 mm from each component and a negligible thickness of approximately 0.1 mm. Measurements were taken with and without the membrane in place. This approach confirmed that the signal captured by the microphone originated from the photoacoustic effect of the membrane in response to the light pulse.

Figure 5 illustrates the setup for the latest characterization test, designed to evaluate the system under conditions closely resembling real-world applications. In this setup, the driving medium was distilled water with a total dissolved solids (TDS) level of less than 1 ppm to minimize electrical noise and undesired induction as the entire system was submerged. A rectangular Pyrex recipient was used to house the setup.

The excitation source was the same LED used in previous tests, secured to one of the recipient’s inner surfaces and sealed with cold silicone. This sealing prevented contact between the LED’s cables and pads with the water, reducing the potential for signal noise. The transducer was positioned at the same height as the LED, aligned with a 30 mm separation.

The sample for this study was a concentrated solution of pure chlorophyll supplied by Trophic. This chlorophyll concentrate was selected due to its spectral absorption characteristics, which closely match the 660 nm wavelength emitted by the LED. Chlorophyll exhibits primary absorption peaks between 430 nm and 452 nm (blue-green region) and a secondary absorption band from 642 nm to 682 nm (red region) [22,23]. These absorption regions have comparable coefficients, enabling efficient utilization of the chlorophyll concentrate under experimental conditions.

The chlorophyll solution was contained within a capillary tube (model WG-1364-1.7) from SP Wilmad-LabGlass (Vineland, NJ, USA). The tube, made of Pyrex, was 100 mm long with an outer diameter of 1.7 mm and an inner diameter of 1.3 mm. It was positioned 5 mm from the LED at the midpoint of the liquid column, ensuring the light beam maximally illuminated the tube containing the sample.

## 3. Results

The initial test, conducted prior to evaluating the photoacoustic effect, aimed to validate the operation of the avalanche oscillator. The circuit, shown in Figure 1, was assembled to determine the transistor’s actual breakdown voltage. At 159 V DC, the transistor exhibited erratic oscillations, indicative of near-threshold behavior. Stable oscillations were achieved at 160 V DC, with an average frequency over a second of 2.79 kHz and a pulse width of 22 ns (10% to 90% amplitude). The output waveform reached a peak voltage of 142 V DC, confirming stable oscillator performance under higher voltage conditions.

These results validate the oscillator circuit’s reliable operation and its capability to generate high-intensity pulses with short durations, demonstrating its suitability for the intended application. Subsequent tests were conducted using a 150 V DC power supply, with the transistor discharge controlled by short pulses. To improve the control over the generated pulses, an external 1 V pulse was used. This external pulse allows for more precise control of the pulse generation, enabling easier management of the averaging process, metrics, and post-processing. By synchronizing the pulse with the system, control over the oscillation frequency and data consistency is significantly improved, facilitating more accurate analysis and measurements.

The subsequent test aimed to evaluate the pulse duration generated by the oscillator at a supply voltage of 150 V DC. A burst of 50 pulses of 1 V DC was used to directly discharge the capacitors into the LED, and the output signal was subsequently averaged. Figure 6 presents two graphs: Figure 6a shows the measurement at the oscillator output with no LED connected and only a 50-ohm resistor to ground.

In this configuration, the pulse duration was 39.3 ns with a standard deviation (STD) of 0.51 ns, equivalent to a variation of 1.3%, with a peak voltage of 137 V with an STD of 2.94 V. As expected, the pulse duration was longer due to the absence of an LED load at the oscillator output. Figure 6b shows the results of the same test with the LED connected to the oscillator output, as configured in Figure 1. In this case, the pulse duration was 19.5 ns with an STD of 0.57 ns, equivalent to a 2.9% variation, with a peak amplitude of 134 V with an STD of 5.56 V. Additionally, the discharge phase of the capacitor, influenced by the small capacitance of the LED, was observed after the electrical pulse ended.

Considering these specifications, the corresponding energy per pulse was calculated to be approximately 13.37 µJ. This parameter is critical for evaluating the system’s photoacoustic signal generation and facilitates direct comparisons with previously reported LED-based photoacoustic sources.

This test demonstrates that a simple avalanche oscillator can excite a high-power LED for durations under 50 ns, delivering a current of 50 A and more than 130 V. This provides sufficient energy to emit a short, high-intensity light pulse.

While the results confirmed the generation of short, high-amplitude electrical pulses, one limitation of LEDs in photoacoustic systems is their relatively slow transition from being completely off to reaching peak intensity, as well as their recovery time. To address this, the subsequent test measured the duration of the light pulse emitted by the LED. The same LED model was used, positioned as illustrated in Figure 3.

Figure 7 shows the results obtained from the optical pulse measurement. Figure 7a shows the response of the LED functioning as a sensor, represented by the green line, while the red line corresponds to the electrical pulse supplied to the other LED serving as the emitter. Figure 7b presents the same signals but with the blockage placed between the LEDs. Note that LEDs exhibit similar rise and discharge times. For practical purposes, the duration of the measured electrical pulse can be approximated as half of the generated optical pulse.

In Figure 7b, a slight disturbance is observed during the electrical pulse, attributed to the electromagnetic pulse generated by the short, high-intensity pulse. This disturbance is unrelated to the emitted light, as the LEDs are optically isolated. In Figure 7a, the signal obtained from the TIA peaks at 0.54 V with a duration of 71.5 ns with a 2.12 ns STD equivalent to a 2.9% variation. Given that both LEDs exhibit similar electrical properties, the emitted light pulse can be estimated to have a duration of approximately 36 ns—roughly double the duration of the electrical pulse. While the LED’s response is slower than that of a laser, it is sufficiently short for photoacoustic system applications.

After confirming the light pulse duration within the range suitable for such systems, a preliminary evaluation was conducted to assess whether the emitted light intensity was sufficient to excite a material compatible with the wavelength used. For this test, the setup shown in Figure 4 was employed, enabling the precise measurement of acoustic signals generated by light pulses incident on the membrane. The acoustic pulses were captured by a microphone within the casing.

Figure 8 presents the results of these tests with and without the membrane, conducted under the same parameters as the pulse duration tests, to ensure consistency in system evaluation. In Figure 8a, the membrane was not included in the test, and the red line represents the electrical signal from the oscillator, while the green line shows the TIA output voltage from the microphone. A short-duration peak with an amplitude of up to 1.6 V is observed, caused by electromagnetic interference during the electrical pulse. However, this peak is unrelated to the recorded acoustic signal. As expected, no acoustic response is observed, as the light intensity was insufficient to excite any casing material at that distance, and no membrane material was present.

Figure 8b test results are shown with the membrane present. The green curve represents the acoustic response captured by the microphone following the light pulse that excited the membrane within the casing. The signal exhibits a maximum amplitude of 0.55 V and a frequency of approximately 1.55 kHz, which depends on the mechanical properties of the membrane and casing. The observed response is a damped acoustic wave, consistent with expected behavior.

These results demonstrate that the LED, when driven by the oscillator, emits sufficient light intensity to excite the membrane and generate a detectable acoustic pulse. This finding highlights the potential of LEDs as a simple and effective method for producing acoustic signals without requiring complex instrumentation, validating the system’s operation.

The final experiment aimed to assess the luminous intensity emitted by the LED to excite a sample submerged in water. Water was chosen as an effective medium for simulating photoacoustic wave behavior and mimicking real-world conditions. Given the speed of sound in water at approximately 1497 m/s, the distance between the excited sample and the ultrasonic transducer was calculated accurately.

Figure 9 illustrates the measurements obtained at the TIA output, which monitored the ultrasonic transducer’s detected signals. Figure 9a shows the full duration of the measurement, including the initial emission pulse, which generates electrical noise captured by the system. This range also importantly encompasses the subsequent acoustic signal detected by the transducer after the LED light pulse excited the sample.

The average acoustic signal detection time was approximately 13.25 µs, consistent with the calculated distance between the sample and the transducer. This confirms that the LED’s luminous intensity was sufficient to excite the sample and generate an ultrasonic pulse detectable by the transducer across a water medium.

The detected photoacoustic signal in water shown in Figure 9b exhibits a maximum amplitude of approximately ±4.0 mV, which presents a significant challenge in terms of the signal-to-noise ratio (SNR). The SNR is a critical parameter that defines the system’s ability to distinguish the useful signal from background noise. In this case, the maximum amplitude of the detected signal is ±4.0 mV, while the background noise, measured under similar conditions, shows a maximum amplitude of ±1.2 mV. The SNR is calculated as the ratio of the maximum signal amplitude to the maximum noise amplitude, resulting in an SNR of approximately 3.33 (10.5 dB). This ratio is relatively low, indicating that the signal can be susceptible to noise interference, especially considering the high gain of the TIA used in the system.

To enhance the SNR and mitigate the impact of noise, signal averaging was employed. With 50 samples averaged per measurement and an LED pulse rate of 100 Hz, the SNR improves, though at the cost of a reduced system refresh rate, which is limited to 2 Hz. While this refresh rate is significantly lower than typical commercial systems, it is necessary to attenuate the noise and improve the reliability of photoacoustic signal detection. It is important to note that this trade-off between SNR improvement and temporal resolution has significant implications for the system’s ability to capture rapid dynamic changes in the signal. The reduced refresh rate affects the temporal resolution and, consequently, the imaging depth. However, this challenge could be addressed through advancements in amplifier design, improved cable shielding, and more effective electrical noise isolation. These optimizations would reduce the need for extensive averaging, allowing for higher sampling rates without sacrificing SNR and thereby improving both temporal resolution and imaging depth, ultimately leading to more precise and reliable photoacoustic measurements.

## 4. Discussion

The results of this study demonstrate the feasibility of employing a simple and cost-effective avalanche oscillator to generate electrical pulses capable of driving a high-power LED to produce light pulses suitable for PAI systems. Unlike conventional PAI setups that rely on Nd:YAG lasers, which remain the industry standard, or alternative LED-based approaches that utilize laser pulse generators and advanced pulse-shaping electronics, this work eliminates the need for such costly and complex equipment. The avalanche oscillator directly produces short-duration pulses without requiring external pulse-shaping circuits, reducing overall system complexity and improving portability while maintaining effective optical excitation for PAI applications.

Table 1 summarizes various studies that have investigated alternative devices to Nd:YAG lasers, primarily employing laser diodes and LEDs. However, a key distinction of those works is their reliance on sophisticated pulse-generation hardware, which significantly increases cost and limits accessibility. In contrast, the approach presented in this study offers a more compact and economical solution, utilizing an avalanche oscillator to generate ultra-short pulses directly from a simple electronic circuit.

The wavelengths utilized in the referenced studies range from 405 nm to 940 nm, depending on the optical properties of the biological tissues under investigation. The proposed system employs a 660 nm LED, well within the biologically relevant range, and its driving circuit can be readily adapted for alternative LED wavelengths, allowing optimization for specific imaging applications.

Pulse durations reported in these studies range between 35 ns and 200 ns. The avalanche oscillator developed in this work generates pulses as short as 19.5 ns, demonstrating its capability to produce even shorter pulses than those typically used. This flexibility allows for an increase in emitted power by easily adjusting the pulse duration. This shorter pulse width allows for increased peak optical power, potentially improving imaging contrast and penetration depth. Additionally, the system provides a level of tunability, enabling adaptation to specific diagnostic needs without the need for complex external pulse-shaping electronics.

Sampling averages in these studies vary significantly, from as few as 20 samples to as many as 4000, reflecting trade-offs between signal-to-noise ratio and temporal resolution. Sample rates reach up to 40 kHz, demonstrating the capabilities of advanced optical diagnostic equipment. The avalanche oscillator developed in this work operates at a frequency of 2.79 kHz in a closed-loop configuration. While this frequency is lower than some commercial LED-based systems, the design allows for straightforward adjustments via circuit modifications or external control signals, making it highly adaptable to diverse application requirements.

Furthermore, the ultrasonic transducer used in this study operates at 2.4 MHz, which is within the range reported in the literature (2.2 MHz to 10 MHz). While some advanced systems utilize transducers up to 20 MHz, the selection of 2.4 MHz ensures a balance between resolution and cost-efficiency, reinforcing the practicality of the proposed system for point-of-care and portable applications.

Table 1 highlights the potential of the proposed device as a viable alternative to more sophisticated and expensive systems. Unlike prior LED-based PAI systems, which typically require external laser pulse generators, this work achieves comparable pulse characteristics with a much simpler and more cost-effective approach. Moreover, the system’s modularity opens opportunities for integrating multiple LEDs to further enhance optical power output, expanding its applicability across various PAI modalities. This research establishes a foundational step toward the development of highly accessible and scalable LED-based PAI systems, paving the way for future optimizations and broader adoption in biomedical imaging.

The slower response time of LEDs compared to lasers inherently affects both imaging speed and spatial resolution. Given the actual oscillator configuration, the system can operate at a sustained frequency of 2.79 kHz in closed-loop mode. Assuming the same averaging of 50 samples, this would yield a final imaging frame rate of 55 Hz. Furthermore, the system configuration could be optimized to further increase this frequency. However, such optimizations fall beyond the scope of this work.

In terms of spatial resolution, the system employs a 2.4 MHz transducer, which sets a theoretical axial resolution of approximately 320 μm in water, calculated based on the acoustic wavelength. The lateral resolution is primarily governed by the LED’s illumination profile and the numerical aperture of the transducer, leading to an estimated resolution in the sub-millimeter range. While these values indicate a modest compromise in resolution compared to laser-based systems, they align with the goal of developing a cost-effective and portable alternative for photoacoustic imaging applications. Future optimizations, including narrower LED pulse widths and improved detection techniques, could further enhance the system’s imaging performance.

The broader spectral bandwidth of LEDs presents both challenges and opportunities for photoacoustic imaging. While it may introduce spectral crosstalk between chromophores, potentially reducing imaging specificity, it also enables the possibility of multispectral imaging without requiring multiple narrowband laser sources. Crosstalk can arise not only from the LED’s broad emission spectrum but also from overlapping absorption spectra of different chromophores, variations in tissue composition, and optical scattering effects that mix signals from adjacent regions. Additionally, acoustic crosstalk can occur due to limited spatial resolution, where signals from multiple absorbers within the ultrasound focal region become indistinguishable. Achieving effective spectral separation would necessitate additional hardware, such as bandpass optical filters or software-based spectral unmixing algorithms to isolate specific chromophores. Regarding real-world applications, tissue scattering remains a key limitation, as the LED’s lower peak power reduces penetration depth compared to lasers, making deep-tissue imaging more challenging.

For future biomedical applications, it is essential to consider safety aspects such as LED power limits and potential thermal effects on tissues. While the current work utilizes synthetic materials for testing, transitioning to biological samples introduces additional considerations. The power of the LED, specifically in terms of its irradiance and the resulting tissue heating, must be carefully controlled to avoid thermal damage to surrounding tissues. The use of pulsed LEDs may reduce the risk of thermal accumulation; however, careful calibration is necessary to ensure that the pulse duration and intensity remain within safe limits for biological tissues. Lasers, which are often employed in more advanced systems, can offer significantly higher power levels than LEDs, raising further concerns about the potential for tissue damage if not carefully managed. Furthermore, the ethical implications of using photoacoustic imaging in biological tissues should be acknowledged, particularly with regard to tissue exposure to optical radiation. Establishing clear safety protocols for optical power limits and pulse parameters, as well as conducting in vivo safety assessments, will be important as the technology progresses toward clinical applications. However, these considerations are beyond the scope of this work, which focuses on the development of the underlying technology and system performance. Addressing these concerns will ensure the safe integration of this system into biomedical diagnostics while maintaining its effectiveness in non-invasive imaging.

## 5. Conclusions

This work has successfully demonstrated the feasibility of utilizing an avalanche oscillator to drive high-power LEDs for photoacoustic imaging applications. By leveraging the avalanche breakdown phenomenon, the oscillator generated high-voltage pulses with durations suitable for exciting LEDs, which subsequently emitted intense light pulses capable of inducing measurable photoacoustic effects.

The experimental results validated the effectiveness of the proposed LED-based photoacoustic imaging system. The system demonstrated the ability to produce detectable photoacoustic signals in both air and water environments, confirming its potential for diverse applications. The use of distilled water as a medium allowed for accurate simulation of real-world conditions.

Compared to traditional laser sources, LEDs offer several key advantages, including lower cost, compact size, and long operational lifetimes. While LEDs exhibit slower response times and broader spectral bandwidths than lasers, the findings show that they are sufficiently capable of generating high-intensity light pulses for inducing photoacoustic responses. These attributes make LEDs particularly attractive for portable and point-of-care imaging systems, where affordability and ease of integration are important.

This work underscores the potential of combining cost-effective components, such as avalanche oscillators and high-power LEDs, to develop photoacoustic systems with practical utility. These findings pave the way for the broader adoption of LED-based photoacoustic imaging technologies in applications ranging from biomedical diagnostics to underwater acoustics. Future work could focus on improving signal-to-noise ratios, enhancing system isolation to reduce electromagnetic interference, and increasing refresh rates through optimized averaging techniques and transducer designs.

## Figures and Tables

**Figure 1 sensors-25-01643-f001:**
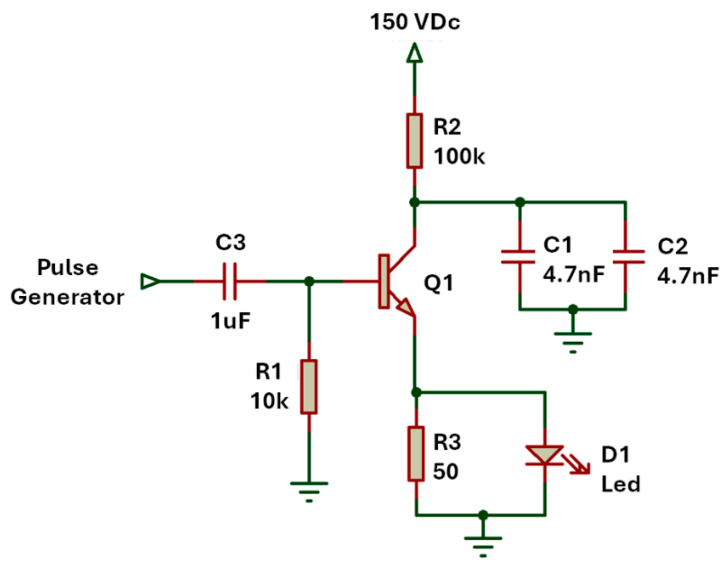
Diagram of the avalanche oscillator with LED array.

**Figure 2 sensors-25-01643-f002:**
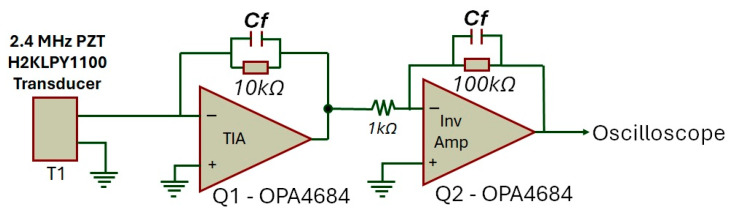
Schematic of the amplification circuit.

**Figure 3 sensors-25-01643-f003:**
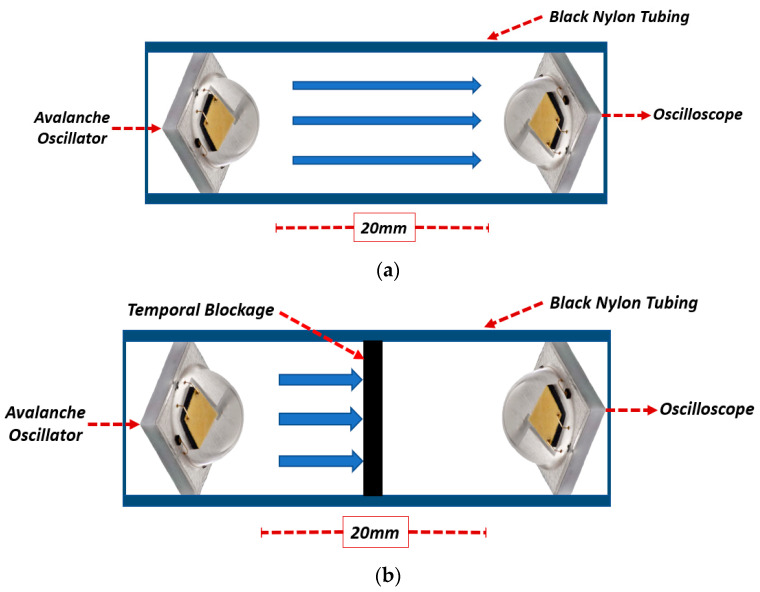
(**a**) Light pulse duration measurement setup without temporal blockage; (**b**) light pulse duration measurement setup with blockage.

**Figure 4 sensors-25-01643-f004:**
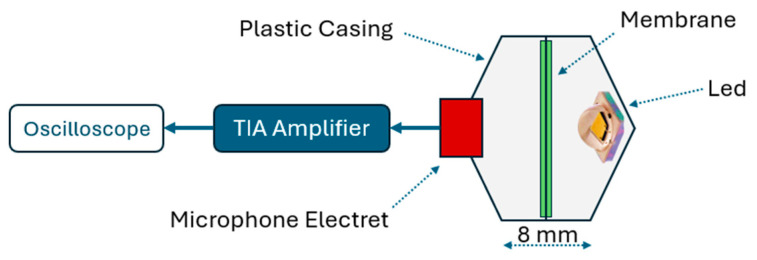
Experimental setup showing the LED, microphone, and black polyethylene membrane in a 3D-printed PETG casing for photoacoustic signal detection in air.

**Figure 5 sensors-25-01643-f005:**
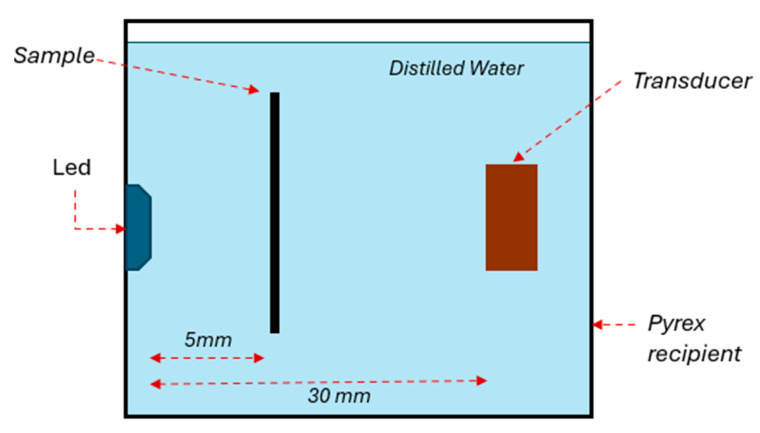
Experimental setup for system characterization in distilled water, showing the LED, transducer, and chlorophyll sample in a Pyrex recipient.

**Figure 6 sensors-25-01643-f006:**
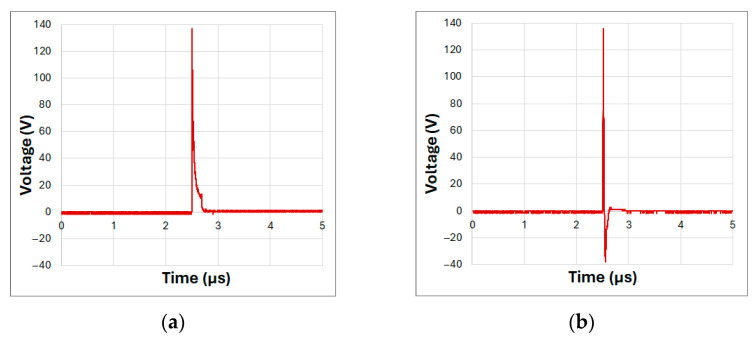
Oscillator output pulse measurements: (**a**) without LED connected, using a 50-ohm resistor; (**b**) with LED connected to the oscillator output.

**Figure 7 sensors-25-01643-f007:**
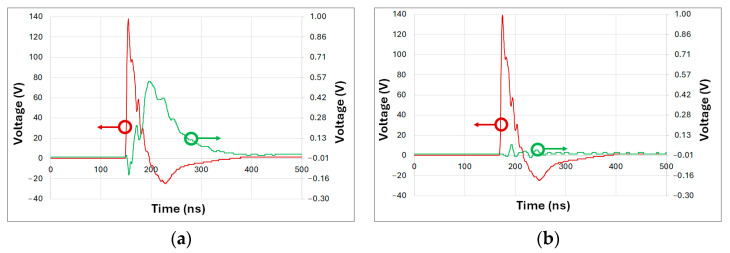
Comparison of electrical and optical pulse measurements. Electrical pulse supplied to the LED (red) and corresponding light pulse measured by the TIA (green). (**a**) Pulse duration measurement without blockage, and (**b**) pulse duration measurement with blockage.

**Figure 8 sensors-25-01643-f008:**
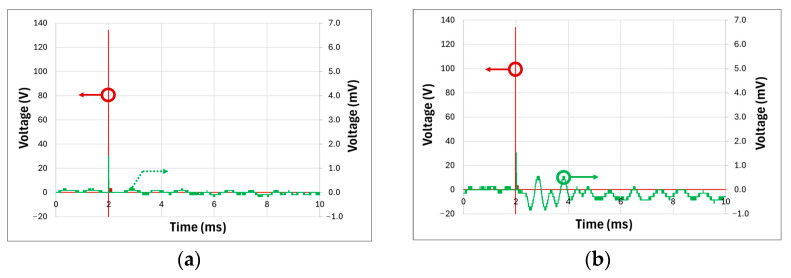
Acoustic signal evaluation: (**a**) TIA output without a membrane, showing no acoustic response (green) from the electrical excitation signal from the oscillator (red); (**b**) TIA output with a membrane, showing a detectable acoustic response (green) following the electrical excitation pulse (red).

**Figure 9 sensors-25-01643-f009:**
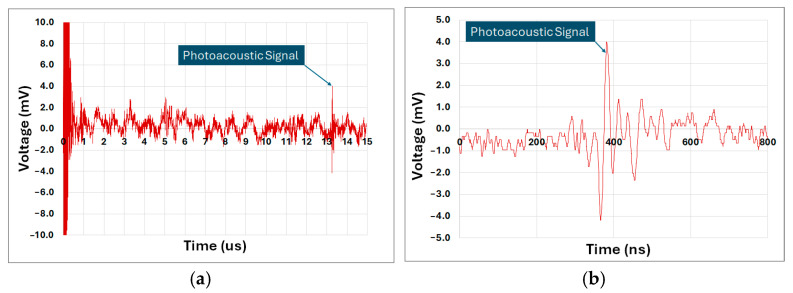
(**a**) Photoacoustic signal measured in water; (**b**) zoomed-in view of the photoacoustic signal measured in water.

**Table 1 sensors-25-01643-t001:** Alternative works about photoacoustic systems.

Ref.	Wavelength	Pulse Duration	Sample Rate	Average	Transducer
[24]	850 nm	70 ns	4 KHz	64	7 MHz
[25]	750 nm	70 ns	4.2 KHz	20	7.1 MHz
[8]	850 nm	100 ns	4 KHz	384	10 MHz
[26]	850 nm	35 ns	1.5 KHz	200	7 MHz
[27]	940 nm	70 ns	4 KHz	640	7 MHz
[28]	905 nm	100 ns	800 Hz	128	4.53 MHz
[29]	580 nm	200 ns	500 Hz	1000	3.5 MHz
[30]	405 nm	200 ns	40 KHz	4000	2.25 MHz
T.W.	660 nm	19.5 ns	100 Hz	50	2.4 MHz

## Data Availability

Data are contained within the article.

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
