# Peer review of "Cost-Effective Photoacoustic Imaging Using High-Power Light-Emitting Diodes Driven by an Avalanche Oscillator"

_sensors, 2025, doi:10.3390/s25061643_

Round 1
Reviewer 1 Report
Comments and Suggestions for Authors
The manuscript presents a cost-effective photoacoustic imaging (PAI) system utilizing high-power LEDs driven by an avalanche oscillator. The study addresses a relevant challenge in PAI by replacing expensive lasers with LEDs, emphasizing portability and affordability. The authors demonstrate the feasibility of generating short, high-intensity light pulses (~19.5 ns) using an avalanche oscillator, successfully detecting photoacoustic signals in air and water. The experimental design, including component specifications and characterization methods, is detailed, and the results align with the stated objectives. While the work highlights the potential of LED-based systems, limitations such as lower power, broader spectral bandwidth, and noise susceptibility are acknowledged. The study contributes to advancing affordable PAI solutions but requires improvements in technical clarity, methodological rigor, and contextualization within the broader literature.
The manuscript shows promise but requires substantial revisions to address critical technical gaps enhance methodological transparency, and strengthen the discussion. Below are detailed recommendations for improvement.
- Figures 6-9 lack statistical validation. Multiple trials should be conducted to confirm pulse consistency, and standard deviations or confidence intervals must be reported. Additionally, the oscilloscope settings (e.g., sampling rate, bandwidth limits) used for pulse measurements are not specified, raising concerns about signal fidelity.
- Table 1 compares the proposed system with prior works but does not explicitly discuss how this study advances the field. Highlight quantitative improvements over existing LED-based PAI systems.
- The detected photoacoustic signal in water (Figure 9b) has an amplitude of +4 mV, which is extremely low, The manuscript does not quantify SNR or propose strategies to mitigate noise. Please include SNR calculations and discuss practical implications for imaging depth/resolution.
- While the slower response time of LEDs compared to lasers is mentioned, the manuscript does not quantify how this affects imaging speed or resolution. Provide metrics such as maximum achievable frame rate or spatial resolution under the current setup.
- The broader spectral bandwidth of LEDs is noted as a limitation, but its impact on imaging specificity (e.g., crosstalk between chromophores) is not analyzed. Discuss whether multispectral imaging is feasible with the current system or requires additional hardware/software adjustments.
- The discussion focuses on technical validation but lacks a critical assessment of the system's suitability for real-world applications. Address challenges such as tissue scattering, safety limits for LED power in vivo, and scalability for clinical use.
- The citation style is inconsistent (e.g. incomplete DOIs in References 9, 22). Ensure all references follow Sensors’ guidelines.
- For future biomedical applications, safety aspects (e.g., LED power limits, thermal effects on tissues) must be discussed. While the current work uses synthetic materials, ethical implications of transitioning to biological samples should be acknowledged.
- Define abbreviations (e.g., TIA, AO) upon first use.
- Clarify why the transistor breakdown voltage (160 V) was not utilized in favor of an external pulse (p. 4).
Author Response
PDF added with responses.
Thank you !

Reviewer 2 Report
Comments and Suggestions for Authors
This paper presents the development and characterization of an avalanche oscillator driver for pumping LEDs suitable for photoacoustic applications. The paper is overall well-written and clear, but the novelty of the proposed solution is not clearly demonstrated by the tests. Could the authors discuss this aspect in more depth?
More specifically, when comparing the new system with the literature, the energy per pulse of the LED is never mentioned, although this is an important parameter for photoacoustic imaging. Could the authors add a comment about this point?
Please also check the resolution of the figures, especially Figure 1: the font is very small and the figure has low resolution. Could the authors uniform the figure fonts?
Author Response
PDF added with responses.
Thank you !

Reviewer 3 Report
Comments and Suggestions for Authors
This paper is very valuable for acousto-optic imaging, and the content of the article is also very detail and substantial, but it is recommended to delete the scientific and conceptual fragments in the paper, such as avalanche oscillator, rise time (212 lines) and fall time (213 lines). Nothing else.
Author Response
PDF added with responses.
Thank you !

Round 2
Reviewer 1 Report
Comments and Suggestions for Authors
The authors have addressed concerns comprehensively and demonstrated a commendable effort to improve the manuscript. The revised version now provides robust statistical validation for Figures 6–9, clarifies technical details (e.g., oscilloscope settings, SNR analysis), and significantly strengthens the discussion by contextualizing the system’s performance against prior works and addressing real-world applicability. The inclusion of metrics such as frame rate, spatial resolution, and safety considerations enhances the manuscript’s rigor and relevance.
While the added SNR analysis is valuable, the manuscript could briefly discuss potential hardware/software strategies (e.g., advanced noise suppression circuits, machine learning-based denoising) to mitigate the low SNR in future iterations.
Overall, the revisions have elevated the manuscript’s quality, and the work represents a meaningful contribution to advancing accessible photoacoustic imaging technologies.